# Hierarchical Porous Heteroatoms—Co-Doped Activated Carbon Synthesized from Coconut Shell and Its Application for Supercapacitors

**DOI:** 10.3390/nano12193504

**Published:** 2022-10-07

**Authors:** Rui Liu, Jing-Xuan Wang, Wein-Duo Yang

**Affiliations:** 1Center of Pharmaceutical Engineering and Technology, School of Pharmacy, Harbin University of Commerce, Harbin 150076, China; 2Department of Chemical and Materials Engineering, National Kaohsiung University of Science and Technology, Kaohsiung 80778, Taiwan

**Keywords:** biomass, hierarchical, porous activated carbon, doped, supercapacitor

## Abstract

Coconut husk biomass waste was used as the carbon precursor to develop a simple and economical process for the preparation of hierarchical porous activated carbon, and the electrochemical properties of the electrode material were explored. The important process variables of carbonization, the weight ratios of the coconut shell/KOH, the amount of source dopant, and the carbonization temperature were investigated in order to reveal the influence of the as-obtained microporous/mesoporous/macroporous hierarchical porous carbon materials on the powder properties. Using a BET specific surface area analyzer, Raman analysis, XPS and SEM, surface morphology, pore distribution and specific surface area of the hierarchical porous carbon materials are discussed. The results show that the as-prepared N-, S- and O-heteroatom-co-doped activated carbon electrode was manufactured at 700 °C for electrochemical characteristics. The electrochemical behavior has the characteristics of pseudo-capacitance, and could reach 186 F g^−1^ at 1 A g^−1^ when measured by the galvanostatic charge–discharge (GCD) test. After 7000 cycles of the charge–discharge test, the initial capacitance value retention rate was 95.6%. It is predicted that capacitor materials made when using coconut shell as a carbon source will have better energy storage performance than traditional carbon supercapacitors.

## 1. Introduction

Supercapacitors (SCs) with high power density have the advantages of short charging time and long service life, which can improve commonly used energy storage devices such that they are able to meet the needs of fast-developing electronic devices. To date, two types of SCs have been developed, namely, electric double layer capacitors (EDLC) based on carbon substrates and pseudo-capacitors based on metal-like oxides [1,2]. The former is operated through the adsorption of ions onto electrode materials with high surface area; and the latter is through the redox Faradaic reaction to store charges. However, because the transfer charge of EDLC is about 100–200 F g^−1^, it is difficult to improve the energy density [3]. In contrast, the proposed capacitors can provide higher specific capacitance (200–1000 F g^−1^) than EDLC, but the conductivity of the electrode material is poor, and it has poor charge–discharge rate capability and cycle performance [4].

The energy accumulation of carbon electrodes depends on the electric double-layer charge on the surface of the material, which limits the specific capacitance of carbon materials. Research combining EDLC and pseudo-capacitance composite materials in order to improve material capacitance demonstrates great potential for development [5]. Metal oxides, such as RuO_2_·nH_2_O [6], CoO_x_ [7], Mn_x_O_y_ [8,9], NiO [10], tin dioxide [11] and TiO_2_ [12] received significant attention because they possess high theoretical capacitance. In particular, the composite material manganese oxide has the advantages of low cost, environmental friendliness, and safety, and is considered to be the most promising material for pseudo-capacitors [13].

The specific capacitance value of the electrode material of the supercapacitor can be determined on the basis of the specific surface area, pore size, pore size distribution, and functional groups on the surface of the material of the electrode [14]. The carbon electrode materials used in supercapacitors include activated carbon [15], carbon nanotubes [16], carbon-derived carbon [17], carbon aerogels [18] and graphene [19]. Supercapacitors with stable cycling properties and fast charging and discharging performance can be prepared, but the optimal pore size distribution and specific surface area of the carbon electrode material cannot be accurately controlled, affecting the conductivity and capacitance of the electrode material. Therefore, in addition to the larger specific surface area of the electrode material, its internal pore size and structure should also be suitable for the adsorption of electrolyte ions, and the electrode surface area must maintain a certain degree of wettability with the electrolyte, which improves the effective utilization of the ionization of the electrolyte, increasing the specific capacitance value of the electrode [20,21]. Activated carbon has become the most widely used electrode material for supercapacitors due to its large specific surface area, high conductivity, high stability and low cost [22,23].

Hierarchical porous carbon includes interconnected micropores, mesopores and macropores. Micropores can promote effective electrode/electrolyte electric double-layer capacitance properties (EDLC); mesoporous channels can improve electrolyte diffusion [24]; and macropores act as electrolyte reservoirs, which can shorten the ionic diffusion distance [25]. The preparation of hierarchical porous carbon materials often employs template technology [26,27] and activation processes [28,29]. Template technology requires the use of a large number of inorganic or organic templates, and the process of template preparation and removal is complicated. Activation can provide porous carbon with a high surface area, but consumes a large amount of activator. For example, KOH [30], NaOH [31], silica [32] and MgO [33] are used in the manufacturing process of activated carbon, and subsequent treatment is required. The generated acid and lye cause damage to the environment [34]. It has been indicated that the template material will produce mesopores or macropores, and activation will produce micropores; the produced hierarchical porous carbon materials have better capacitance than microporous carbon, and provide a better choice for energy storage devices [35].

A series of studies have explored the application of activation or template technology in the synthesis of porous carbon with a micropore size. The choice of alkalis with different metal ions can be used to accurately control pore size [36]. Microporous porous carbon can increase the specific capacitance, so it is very important to prepare hierarchical porous carbon with a large number of different micropore sizes [37,38]. Due to the adjustable pore size of porous carbonaceous material, it can be used to create a path of access for ion penetration, and is considered to be an ideal material for supercapacitor electrode materials [39]. In addition, porous carbon materials are usually synthesized using the carbonization activation method, and the carbonization process is simple and low cost [40,41].

A variety of carbon precursors have been developed to produce carbon materials, such as ionic liquid [42], biomass [43,44] and polymer [45], and these are used as raw materials. Meanwhile, it can also be synthesized as doped carbon material on-site. Among these precursors, waste biomass has received extensive attention due to its rich number of sources, low price and environmental friendliness [46]; sources such as onions [47], coffee [48], pomelo peel [49], seaweed [50] and coconut shell [51] have been developed as raw materials for the preparation of activated carbon materials.

Coconut shell is mainly composed of cellulose, hemicellulose, and lignin. It grows in tropical and subtropical regions. Therefore, research into applying coconut shells as biomass raw material for the preparation of porous carbon materials demonstrates significant development potential [52,53]. Coconut shell is mainly composed of cellulose, hemicellulose and lignin. It grows in tropical and subtropical regions. Lee et al. [54] used coconut shell-derived activated carbon for the preparation of high-performance solid-state supercapacitors. The ratio of KOH to hydrochar was changed in a systemic way to study how it influences the structure and electrochemical behavior of the capacitor. Jain et al. [55] prepared activated carbons derived from coconut shells as high-energy-density cathode materials for Li-ion capacitors. It was indicated that a dramatic increase in energy density of ~69 Wh kg^–1^ and an extraordinary cyclability of ~ 2000 cycles could be obtained in the capacitors by employing a tailored mesoporosity of ~ 60% in the coconut shells. Furthermore, the use of coconut shell porous carbons with a tunable micro-/mesopore ratio in the preparation of high-performance supercapacitors was also investigated by Mi et al. [56]. Granular micro-/mesoporous carbon with a ratio of mesopore to total pore volume (V_meso_/V_total_) greater than 75% was prepared using coconut shells as a precursor by means of a one-step thermal treatment. Electrochemical examinations indicated that, with increasing V_meso_/V_total_ of the porous carbons, the equivalent series resistance (ESR) decreased, resulting in a capacitance retention of 93% at 5 A g^−1^. Yin et al. [57] also indicated that three-dimensional hierarchical porous activated carbon could be obtained from coconut fibers via KOH activation, and exhibited high-rate performance for symmetric supercapacitors.

The specific capacitance of the pseudo-capacitor material demonstrates better capacitance and energy density performance due to the charge storage mechanism of the redox reaction on the surface of the material. In these pseudo-capacitor materials, N, P, S, O and other atoms can be added through the high-specific-surface-area nanostructure of EDLC [58,59,60,61] to form heteroatoms in addition to the original EDLC structure of the material. The doping can cause additional pseudo-capacitance effects between the ions in the electrolyte and the surface. Since S atoms are larger than N atoms and have low electronegativity, S doping demonstrates better advantages in terms of many characteristics, including increases in interlayer spacing, number of active sites and electrochemical properties [62]. N- and S-doped materials have a special synergy between N and S atoms, and show good electrochemical performance. Nowadays, they have been widely used in sodium ion batteries, Li-ion batteries, electrocatalysis and supercapacitors [63].

Recently, bimetallic oxide nanomaterials combined with carbon materials have had a significant impact on the development of high-performance electrode materials for supercapacitors. Kim et al. [64] reported that hollow carbon nanofibers with inside–outside decoration of Bi-metallic metal–organic framework (MOF)-derived Ni-Fe phosphides could be used as electrode materials for the preparation of asymmetric supercapacitors. Bi-metallic MOF-generated phosphides within and outside the hollow carbon nanofibers (HCNFs) showed remarkable potential for energy storage owing to their improved conductivity and high specific capacitance. Moreover, a self-templated MOF-based strategy for the synthesis and assembly of bimetallic oxides/nanoporous carbon nanostructures (Ni–Fe–O/NPC) on porous carbon nanofibers was also developed. It shows a high specific capacitance of 1419 F g^−1^ at 1 A g^−1^ and good cycling life with capacitance retention of approximately 88.5% after 10,000 cycles [65]. Additionally, in addition to porous active carbon having been used as an electrode material for supercapacitors, cellulose nanocrystal (CNC) nanomaterials demonstrate unique physicochemical properties, low cost, biocompatibility and biodegradability. The performance and processing method for CNC supercapacitors have attracted attention from researchers. MnO/CNC/rGO fibers from colloidal liquid crystal for flexible supercapacitors via a continuous one-process method was reported by Yuan et al. [66]. Flexible supercapacitors based on fiber-shaped electrodes showed high potential for practical application. The synthesis of CNC and manganese oxide nanoparticles in GO aqueous dispersion not only prevented GO nanosheets from restacking, but also ensured a uniform intercalation of nanoparticles. The fiber supercapacitor also demonstrated a good cycling stability. Su et al. [67] developed a facile synthesis of CNC–MnO_2_ hybrid as a supercapacitor electrode. This hybrid was prepared by anchoring porous MnO_2_ nanowhiskers over the surface of CNCs. The electrochemical properties of the CNC–MnO_2_ electrode included high specific capacitance, with values of up to 387 F g^−1^, and good cycling stability. Moreover, the results suggest that the CNC–MnO_2_ hybrid is comparable or superior to other carbon-based materials coated with MnO_2_ whiskers. Recently, Durairaj et al. [68] published a review: “Cellulose Nanocrystals (CNC)-based functional materials for supercapacitor applications”. It places a spotlight on the recent research contributions of novel CNC-conductive materials and CNC-based nanocomposites, as well as their potential applications as supercapacitors (SCs). The synthesis of CNCs was performed using renewable carbon sources. This review also introduced the significant scientific achievements and industrial uses of nanoscale materials and composites for energy conversion and storage applications.

The preparation of porous carbons attracts a great deal of attention, given the importance of these materials in many applications, and more sustainable chemical activation strategies for the production of porous carbons have been developed [69]. Taiwan is located in the subtropical region, and has a high coconut production. Coconut is used as a summer drink and as a source of coconut oil; however, the waste biomass material from coconut shells also poses a large burden on the environment. The resource reuse of coconut shell waste biomass material is a major issue for sustainable development. Research into improving the existing process, shortening the manufacturing process, reducing the carbonization temperature, and reducing the use of harmful chemicals can indeed make some contributions on the sustainable development of the porous activated carbon manufacturing process.

This study aims to address the need for environmental sustainability in the industrial mass production of activated carbons, to study energy saving and simplify their process of manufacturing. Using coconut shell biomass waste abundant in Taiwan as raw material, a single substance as a polyatomic (N, S and O) dopant, a hierarchical porous activated carbon with excellent properties was prepared, which is used as a negative electrode material for supercapacitors. Moreover, to the best of our knowledge, combining relevant multi-faceted study to reveal the relationship between interfacial adsorption capacity of activated carbon electrode and aqueous electrolyte to improve the capacitance of supercapacitors are rare, and worthy of investigation.

In this study, in order to utilize biomass resources, waste coconut shell was used as a carbon precursor to prepare hierarchical microporous/mesoporous/macroporous materials through carbonization. Through the use of S, N and O atom doping, the carbon material was dried via freeze-drying technology, greatly reducing the shrinkage and deformation of the carbon material during the drying process, and resulting in a carbon material with hierarchical pores and excellent pore properties, greatly improving the electrochemical properties for use in supercapacitors.

## 2. Materials and Methods

### 2.1. Materials

Analytical-grade chemicals were used as received, without any further purification. Coconut shell (from Ping-Tong, Taiwan), potassium hydroxide (98% purity, Showa, Tokyo, Japan), HCl (Showa, Tokyo, Japan) and ammonium persulfate (98% purity, Honeywell, Hannover, Germany) were utilized to prepare activated carbon powder. Nickel foam (NF) (98% purity, Fluka, Tokyo, Japan), carbon black and polyvinylidene fluoride (PVDF) (Showa, Tokyo, Japan) was used to prepare AC/NF electrode. Sodium sulfate (≥98% purity, Honeywell, Hannover, Germany) was used as the electrolyte.

### 2.2. Preparation of Hierarchical Porous Activated Carbon

Coconut shell was dried in an oven at 105 °C for 24 h, and pulverized into fine powder; then, the powder was sieved. The powder that was between 20 and 100 mesh was obtained and stored in a refrigerator at 4 °C.

In this experiment, the so-called undoped activated carbon (AC) material was prepared in advance. Ten grams of dried coconut shell powder was mixed with different weight ratios of KOH (weight ratios of coconut shell/KOH at 1/1~1/5), placed into a tube furnace, and heated under an environment with an N_2_ flow rate of 50 mL min^−1^ with a rate of temperature increase of 10 °C min^−1^ up to 700 °C, and kept for 1 h to perform high-temperature carbonization. After cooling, it was washed with 1 M HCl to remove impurities, followed by washing with DI water to obtain a pH value of approximately 7, and then the obtained hierarchical porous carbon material was prepared by freeze drying.

For the preparation of heteroatom-doped (S, N and O element-doped) hierarchical porous carbon material, 10 g of coconut shell powder, KOH and ammonium persulfate at various weight ratios were also mixed in. For the preliminary experiment of synthesizing activated carbon doped with heterogeneous atoms, the one-step heat treatment and activation process was performed by means of the mixture of KOH with ammonium persulfate and then coconut shell powder. At 750 °C, the coconut shell was turned into ashes, and the activated carbon could not be obtained because the protective gas (N_2_) at a flow rate of 50 mL min^−1^ is not enough to be introduced for the preparatory conditions at the high temperature above 750 °C. In this case, under the synergistic effect of KOH and ammonium persulfate, the produced porous activated carbon could not be maintained, and then reacted into CO and CO_2_. It was indicated that both thermal and chemical activation of persulfate can be formed SO_4_^−^, which is a very powerful oxidant [70]. KOH is an activator. During the activation process, it is first decomposed into K_2_O and H_2_O, and then reacts with the produced activated carbon to form CO or CO_2_ [71]. The higher the temperature, the more intense the reaction. Therefore, activation studies were performed at temperatures below 700 °C in this study. The as-prepared mixture was placed in a tube furnace, and treated in a manner the same as the above undoped AC at a temperature of 400–700 °C in order to perform carbonization at a range of different temperatures. After cooling, it was washed with 1 M HCl to remove impurities, followed by washing with DI water to obtain a pH value of approximately 7, and then dried in a freeze dryer to obtain heteroatom-doped hierarchical porous carbon material.

The prepared undoped AC sample is represented by C/KOH/x, where x is the weight multiple of KOH to the coconut shell; in contrast, the prepared doped AC is represented by C/KOH/S-y:z, where S is the doping source (ammonium persulfate), and y:z is the weight ratio of KOH to ammonium persulfate in the raw materials. In this study, the activation was generally performed at a temperature of 700 °C; however, in order to explore the effect of the activation temperature, some doped ACs were activated at different temperatures between 400 and 700 °C. In the samples where the activation temperature was different from the general activation temperature (700 °C), the activation temperature has been included in the sample name. The procedure for preparing AC from coconut shell is shown in Figure 1. Table 1 shows the preparatory conditions of the as-prepared activated carbon obtained using coconut shell as a raw material.

### 2.3. Characterizations

A field-emission scanning electron microscope (FESEM, JEOL6330, Tokyo, Japan) with an acceleration voltage of 80 kV was utilized to examine the microstructures of the as-prepared micropore/mesopore/macropore hierarchical porous carbon materials. The Brunauer–Emmett–Teller (BET) surface area, Barrett–Joyner–Halenda (BJH) mesopore area, t-plot micropore area and N_2_ adsorption–desorption isotherms were measured with a Micrometrics ASAP 2020 instrument (Micrometrics, Atlanta, Georgia, U.S.). The Raman spectrum was determined using a Jobin–Yvon Lab Ram HR800 (HORIBA Jobin Yvon Inc., Paris, France) Raman spectroscope equipped with a 514.5 nm laser source. X-ray photoelectron spectroscopy (XPS, ULVAC–PHI, PHI 5000, Chigasaki, Japan) patterns were obtained using a monochromatic Al-anode X-ray gun. A binding energy of 284.6 eV for C was used to calibrate the charge-shifted energy scale. The spectra were deconvoluted for chemical identification using 100% Gaussian peaks.

### 2.4. Electrochemical Properties

The mixture of AC materials, polyvinylidene-fluoride (PVDF) as a binder and carbon black as a conductive additive at a weight ratio of 85:10:5 was dispersed in N-methyl pyrrolidone and then mixed for 12 h by magnetic stirring to generate a homogeneous mixture. The resultant slurry was coated onto Ni foam as a current collector.

All electrochemical measurements were carried out in 1 M Na_2_SO_4_ electrolyte. The electrochemical experiment was measured in a conventional three-electrode system, using the as-prepared AC/Ni foam composite (1.0 cm × 1.0 cm) as the working electrode, a Pt wire (1.0 cm × 1.0 cm) as the counter electrode and a saturated calomel electrode (SCE) as the reference electrode.

Cyclic voltammetry (CV) and galvanostatic charge–discharge (GCD) were measured using a CHI 760D electrochemical workstation.

Moreover, the specific capacitance (*C_m_*) was obtained according to the discharge curve of the GCD test using Equation (1):(1)$$C_{m}=\frac{i\times \Delta t}{\Delta V\times m}$$
where *i* (A) is the discharge current, Δ*t* (s) is the discharge time, Δ*V* (V) is the discharge potential difference, and *m* (g) is the mass of the porous activated carbon material.

Electrochemical impedance spectra (EIS) were determined at an open-circuit voltage, with a bias of 10 mV for frequencies ranging from 100 kHz to 0.01 Hz, to test the electron transport properties.

## 3. Results and Discussion

### 3.1. Characterization of Activated Carbon Materials

#### 3.1.1. Examination of Morphological Properties

An SEM image of coconut shell after activation with KOH at 700 °C is shown in Figure 1. It can be observed in the figure that the prepared activated carbon has different morphologies. Figure 1a shows the AC sample obtained from the coconut shell treated without KOH activator (C/KOH-0). The as-obtained activated carbon sample with a weight ratio of coconut shell/KOH of 1.0 (C/KOH-1) shows a spongy structure, indicating successful activation of the sample on the surface; Figure 1b. It can be seen that a small-pore structure has been formed. Furthermore, the pores of the obtained activated carbon increase in size with increasing addition of KOH, as shown in Figure 1c–f. It can be explained that, due to the CO, CO_2_, water vapor and other gases produced in the chemical activation process, many holes are formed in the activated carbon, and the number of these defects increases in the activated carbon.

This study used a BET specific surface to analyze the AC obtained using coconut shell biomass as raw material. Figure 2a shows the nitrogen adsorption and desorption curve of the undoped activated carbon powders. It can be observed that there is no significant difference between the adsorption and desorption curve types for each material. They are all of the first type of adsorption isotherm. The adsorption isotherm increases rapidly at a lower relative pressure. After reaching a certain relative pressure, the adsorption saturation value appears, which is formed by a structure consisting of mesopores with a large number of micropores [72]. The adsorption capacity of activated carbon obtained at a weight ratio of coconut shell/KOH of 1/5 (C/KOH-5) was the greatest, as it had the largest specific surface area. Figure 2b shows the pore size distribution curves of different undoped activated carbon. The results show the pore size of the undoped activated carbon was mostly in the range of 1–10 nm. The pore size and distribution of the AC powder can be judged according to the IUPAC pore type classification. Activated carbon is a material composed of a large number of micropores and mesopores, which is the same as the above-mentioned isotherm adsorption curve.

Figure 3a shows the nitrogen adsorption and desorption curve of heteroatom-co-doped activated carbon. It can be seen that there are also no significant differences between the adsorption and desorption curve types of each material, and they are all of the first type of adsorption isotherm. The structure is composed of mesopores with a large number of micropores. Obviously, the doped AC obtained from coconut shell, KOH and ammonium persulfate in the weight ratio 1:3:1 (C/KOH/S-3:1 sample) had the greatest adsorption capacity, which indicates that it had the highest specific surface area. Figure 3b shows the pore size distribution curve of heteroatom-co-doped AC (C/KOH/S sample). The results indicate that most of the C/KOH/S samples had a pore size in the range of 1–10 nm. According to the IUPAC pore type classification, it can be judged that the activated carbon synthesized in this study is a material composed of a large number of micropores and mesopores, which is the same as the judgment made on the basis of the above-mentioned N_2_ isotherm adsorption curve.

Table 2 presents the results of the analysis of the specific surface area and pore properties of the as-prepared AC. It can be seen from the table that the specific surface area of undoped AC prepared without the use of KOH as activator (C/KOH-0) was 377.7 m^2^ g^−1^. The largest fraction of specific surface area of the as-obtained AC was obtained for a weight ratio of coconut shell/KOH of 1/5 (C/KOH-5), increasing to 2730.7 m^2^ g^−1^. However, for the C/KOH-5 sample, the fraction of micropores decreased to 11.8%, which is due to the fact that the K-containing species from KOH in the chemical activation process of this case can corrode the structure, forming a microporous structure; however, owing to the larger amount of gas generated, the pore structure is destroyed, forming a mesoporous structure. In this study, the C/KOH-3 sample, obtained with weight ratio of coconut shell/KOH of 1/3, had both a high specific surface area and a high micropore ratio, and was used for the subsequent study of the heteroatom-co-doped activated carbon, co-doped AC and the specific surface area and pore properties of doped activated carbon. The specific surface area of the C/KOH/S-3:1 sample was 1587.4 m^2^ g^−1^, which is slightly lower than the value of 1601.8 m^2^ g^−1^ obtained for the previous C/KOH-3 sample (activated carbon without dopant). The specific surface area and the micropore ratio of the doped activated carbon exhibit a downward trend with increasing amounts of ammonium persulfate addition during the decomposition process. Therefore, this study will follow-up on the C/KOH/S sample by performing an electrochemical analysis to determine the parameters with which the best capacitance can be obtained.

#### 3.1.2. Spectroscopic Analysis of Activated Carbon

Activated carbon is used for the capacitance of electrode materials, and the degree of graphitization of activated carbon is a key factor in improving electrical properties [73]. Furthermore, the degree of graphitization of samples can be obtained by means of Raman spectroscopy. Figure 4a shows an analysis of the Raman spectra of the activated carbon materials. From the figure, all samples show two different characteristic peaks, which are centered at 1350 and 1595 cm^−1^, representing the D band and G band of the carbon material, respectively.

The D band is a common feature for sp^3^ defects or disorders in carbon, and the G band provides useful information on in-plane vibration of sp^2^-bonded carbon atoms in a 2-D hexagonal lattice [74]. The intensity ratio between the D band and the G band (I_D_/I_G_) is considered to be an important indicator of amorphous components in carbon materials. The higher the ratio, the lower the degree of graphitization. It can be seen from the figure that with increasing carbonization temperature, the peak value of the G band gradually increases, causing the I_D_/I_G_ value to decrease; therefore, the degree of graphitization is also higher. Among them, the doped AC prepared using coconut shell, KOH, and ammonium persulfate at weight ratio of 1:3:1 and activated at a temperature of 700 °C (C/KOH/S-3:3-700) had the lowest I_D_/I_G_ value, at 0.79, meaning that it possessed the highest degree of graphitization. The C/KOH/S-3:3–700 material was carbonized at 700 °C, which is a temperature that is sufficiently high that the carbon atoms were more regularly arranged, and exhibited the lowest I_D_/I_G_ value by a significant degree. The carbon material had a more orderly and regular arrangement of carbon atoms inside following graphitization by means of high-temperature treatment, along with improved crystallinity, which is beneficial for the conductivity of the material and can effectively improve the capacitance value of the supercapacitor [75].

Figure 4b shows XRD patterns of the as-obtained co-doped porous carbon activated at different temperatures. The patterns exhibit a broad diffraction peak of 2θ at around 24° and 42°, corresponding to the (002) and (100) reflections of graphic carbon with a disorder phase, respectively [76]. The XRD pattern shows that with increasing activation temperature, the (002) broad peak became narrower, indicating that the structure exhibits a more regular arrangement, that is, the degree of graphitization increased. At lower temperatures (500 °C), a small amount of K_2_O was found; at 600 °C, small amounts of K_2_SO_4_ and K_2_CO_3_ were present. This is the result of preparing activated carbon using KOH and ammonium persulfate as activators, and the activator decomposes at high temperature [70]. It is worth mentioning that these impurities hardly appeared following activation at 700 °C, indicating that these small amounts of dopant atoms were attached to carbon atoms, and could not be detected by XRD. For further comparison, the XRD results of the other samples are also shown in the Appendix A. By comparing these with the results presented in Figure 4a, it can be seen that the effect of carbonization temperature on graphitization is consistent with the results of the previous Raman analysis.

In order to study the chemical composition of doped activated carbon materials, XPS analysis was performed and calibrated by C at the binding energy of 284.8 eV. Moreover, the binding energy curves of S, N and O were deconvoluted by fitting a Gaussian curve and using the same type of background line (Shirley). The agreement between the summed spectrum and the raw spectrum was as high as 98% or more. Figure 5a presents a survey of the all-range binding energy determined via XPS analysis for the C/KOH/S-3:3-700 sample. The results confirmed the presence of S, N, O, and C elements on the surface of the sample; C is the main element; S, N and O are heteroatom-co-doped with the porous carbon material.

Figure 5b shows the S 2p spectrum, which can be deconvoluted into four peaks. The four main peaks are centered at 163.6 eV (S4), 164.8 eV (S3), 168.6 eV (S2) and 169.3 eV (S1), respectively. The first two peaks can be attributed to S 2p_3/2_ and S 2p_1/2_ of the thiophene C-S-C covalent bond from spin-orbital coupling. The other two peaks are different sulfur oxide forms of C-SO_x_-C (x = 2~4) [27]. This result indicates that the doping of activated carbon with sulfur atoms was successful. Li et al. [77] reported that thiourea (CN_2_H_4_S) was used as both the N and S precursors to prepare co-doped porous carbon from willow catkin for supercapacitors. The deconvolutional peaks of S 2p in the literature are similar to this study. However, comparing with the S 2p of XPS from the literature without using the O element of the doping source (CN_2_H_4_S) reveals that the as-obtained activated carbon of this study exhibits more relatively strong S1 and S2 deconvolutional peaks, revealing it contains higher C-SO_x_-C in the carbon. It could be explained that the as-prepared porous activated carbon contained a high amount of O element, because (NH_4_)S_2_O_8_ was a dopant source.

In the N 1s spectrum, the four individual peaks located at 398.6 eV (14.0%), 400.1 eV (32.7%), 401.0 eV (21.5%) and 401.8 eV (31.8%) represent pyridinic-N, pyrrolic-N, quaterary-N and oxidized-N, respectively. The pyridinic-N (14.0%), pyrrolic-N (32.7%) and oxidized-N (31.8%) distributed on the material surface and edge are the main providers of pseudo-capacitance [59]. Moreover, the presence of quaterary-N (21.5%) can enhance electron transfer in order to increase conductivity. Therefore, the high N content with surface activity plays an important role in improving the capacitance performance of the carbon material (Figure 5c). The deconvolution peaks of N 1s are also consistent with the results in the literature [43]. Kong et al. studied B, O and N co-doped laver biomass to prepare porous carbon for energy storage. KCl/ZnCl_2_ was as a combined activator and H_3_BO_3_ was used for dopant source. Although, the positions of the four deconvolutional peaks of N 1s in this study are similar to Kong’s report, by comparing those peaks in detail, the carbon oxidized-N of this study exhibits the highest intensity of the four peaks. In contrast, the oxidized-N peak in the above literature is the weakest for the N 1s deconvolutional peaks. This is also due to the fact that the activated carbon prepared by this study contains substantial amounts of of O dopant. Additionally, the core level binding energy of the O 1s spectrum was split into two peaks at 531.8 eV and 533.4 eV, corresponding to C=O (82.5%) and C-CH/C-O-C (17.5%), respectively (Figure 5d). The results reveal that a high ratio of O is bonded to the C atoms; due to the high degree of hydrophilicity of O, the carbon electrode is able to promote the adsorption capacity of the aqueous electrolyte, which is beneficial to increasing the capacitive properties [27].

Moreover, the as-prepared porous activated carbons were also investigated for their hydrophilic properties using a contact angle meter. The results show that the C/KOH/S-3:3-700 sample coated on the glass slide possesses the lowest water contact angle, at approximately 23°. In contrast, the water contact angle of the undoped carbon sample (C/KOH-3) is about 32° (Appendix A). It is speculated that utilizing ammonium persulfate as the doping source, owing to the O, S and N elements possessing higher electronegativity than C; thus, the synergy effect of the three elements can greatly improve the hydrophilicity of the as-obtained porous carbon electrode material, resulting in the high hydrophilicity of heteroatoms co-doped porous carbon electrode significantly promoting the adsorption capacity of the aqueous electrolyte and enhancing the capacitive properties. The adsorption capacity of the electrode surface to the aqueous electrolyte was also confirmed by the subsequent EIS analysis.

Table 3 presents the analysis of the atomic composition of the as-obtained AC samples. It can be seen from the table that the atomic contents of N, O and S in C/KOH/S-3:1 were 4.09%, 9.29% and 2.16%, respectively. In the undoped AC, with increasing fraction of added ammonium persulfate, the sulfur content of C/KOH/S-3:5 AC sample increased to 5.64%. The atomic contents of N, O and S in C/KOH/S-3:3 calcined at 400 °C were 3.15%, 9.29% and 15.49%, respectively. Moreover, with increasing activation temperature, the atomic contents of N, O and S in C/KOH/S-3:3 calcined at 700 °C decreased to 3.88%, 6.78% and 4.76%, respectively.

### 3.2. Electrical Analysis of Activated Carbon

#### 3.2.1. Effect of KOH Weight Ratios

The influence of different KOH weight ratios on the electrochemical performance of the prepared activated carbon electrode was examined by means of cyclic voltammetry (CV) and galvanostatic charging and discharging (GCD) tests.

The CV curves of undoped AC (C/KOH) electrodes at 50 mV s^−1^ prepared with different weight ratios of KOH are shown in Figure 6a. The CV curve of the C/KOH-3 sample possesses a larger area, indicating a better capacitance performance. Figure 6b shows the CV curve of the C/KOH-3 electrode at a scan rate of 10–50 mV s^−1^. The CV curve still has a rectangular shape, indicating that the prepared C/KOH-3 electrode has ideal EDLC characteristics. Figure 6c shows the charge–discharge analysis of undoped AC electrode prepared by adding different weight ratios of KOH at 1 A g^−1^. The specific capacitance of the C/KOH-3 electrode was calculated to be 77.4 F g^−1^ using Equation (1). In combination with the BET analysis, this indicates that the specific surface area of C/KOH-5 is the largest; however, due to the fraction of micropores being too low, the ion storage capacity is low, resulting in a decrease in capacitance.

Thus, in order to obtain electrode materials with high specific capacitance, it is necessary to choose materials with both high specific surface area and a high fraction of micropores. The C/KOH-3 sample meets these requirements, and is the object of the subsequent electrode study. GCD tests were performed at different current densities of 1, 2, 4, 6, 8 and 10 A g^−1^, as shown in Figure 6d. It was calculated that the specific capacitance values of the C/KOH-3 electrode at different current densities of 1, 2, 4, 6, 8, and 10 A g^−1^ were 76.6, 61.7, 54.3, 49.7, 45.7 and 44.3 F g^−1^, respectively. This is because with increasing current density, the ions in the electrolyte only react quickly with the surface of the material, and cannot fully enter the pore structure for charge storage, resulting in a decrease in specific capacitance.

#### 3.2.2. Effect of Ammonium Persulfate as a Dopant Source

The CV curve at 50 mV s^−1^ of the heteroatom-co-doped AC electrode prepared by adding ammonium persulfate in different weight ratios is shown in Figure 7a. The CV curve for the C/KOH/S-3:3 sample (obtained from weight ratio of KOH/ammonium persulfate at 3:3) has a larger area, indicating a better capacitance performance. Figure 7b shows the CV curve at a scan rate of 10–50 mV s^−1^ of the C/KOH/S-3:3 electrode. The electrochemical characteristics of the doped AC demonstrated a better performance than that of the undoped AC. For the doped AC, the electrode was slightly deformed, which can be attributed to the success of the C/KOH/S electrode doping S, N and O atoms into the carbon structure of the C/KOH/S electrode; and it has a rectangular shape, indicating that the prepared N, S co-doped AC of C/KOH/S-3:3 electrode has both EDLC characteristics and pseudo-capacitance characteristics. Figure 7c shows the charge–discharge analysis at 1 A g^−1^ of C/KOH/S electrodes with different weight ratios of ammonium persulfate. The results show that the specific capacitance of the C/KOH/S-3:3 electrode calcined at 700 °C reached as high as 184 F g^−1^.

According to the results of the BET analysis, in the C/KOH/S electrode prepared by adding ammonium persulfate, the specific surface area and the fraction of micropores of the material decreased; however, the doping of S, N and O atoms can cause the material to exhibit pseudo-capacitance characteristics, and to reach a higher level of capacitance. The pseudo-capacitor material doped with N, S and O atoms exhibits a slight curvature in terms of its charge and discharge curve at low current densities. This is caused by the redox reaction inside the material.

The GCD measurements of the C/KOH/S-3:3 electrode calcined at 700 °C at different current densities (1, 2, 4, 6, 8 and 10 A g^−1^) are shown in Figure 7d. It was calculated that the specific capacitance values of the C/KOH/S-3:3 electrode at the different current densities of 1, 2, 4, 6, 8 and 10 A g^−1^ were 184, 160.9, 145.1, 138.9, 133.4 and 129.6 F g^−1^, respectively, on the basis of the figure and using Equation (1). This is because, with increasing current density, the ions in the electrolyte react quickly with the surface of the material, and cannot fully enter the pore structure for storage, resulting in a decrease in the specific capacitance [78]. Among the activated carbon electrodes doped with heteroatoms such as NSO, the sample C/KOH/S-3:3 exhibited the best capacitive properties (by GCD), which may be related to the combination of its SSA and doping amount being the most suitable. Bhattarai et al. [71] reported cherry waste-derived activated carbon (CFAC) with a self-doped N for application in supercapacitors and sodium-ion batteries. It was reported that the specific surface area and nitrogen content were observed to play a very delicate role in determining the charge storage ability of the CFAC, and the performance could be optimized only by carefully balancing both of these properties.

The energy storage of the activated carbon electrode is utilized in the capacitive mechanism of the EDLC; thus, ions are adsorbed onto the surface of the activated carbon electrode immersed in the electrolyte, forming an electric double layer to store the charges. The capacitance is proportional to the area of the electric double layer formed on the activated carbon interface. Because the EDLC demonstrates no chemical reaction between the electrode’s active materials, it can be rapidly charged and discharged, with charging and discharging being a manifestation of ion adsorption. EDLC supercapacitors demonstrate less aging and excellent charge–discharge cycling characteristics. Additionally, the presence of heteroatoms (for example S, B and N) in the porous activated carbon could increase capacitance by providing locations for pseudo-capacitance while also increasing electrical conductivity [59]. Furthermore, the synergistic effect of multi-heteroatom-doped carbon materials on electrochemical performance is superior to that of single heteroatoms [79].

#### 3.2.3. Electrical Analysis of Different Activated Carbon Materials

Figure 8a shows the GCD behavior at 1 A g^−1^ of C/KOH-3, C/KOH/S-3:5, C/KOH/S-3:3-400 and C/KOH/S-3:3-700 electrodes. The specific capacitances are calculated to be 77, 127.7, 11.4 and 184 F g^−1^, respectively. The specific capacitances of the four different electrodes are calculated by GCD, as shown in Figure 8b. Since the diffusion rate of ions is at a lower current density, the ions have sufficient time to enter the pore structure of the activated carbon for storage. Therefore, higher capacitance can be obtained at a lower current density compared with at a higher current density.

Electrochemical impedance spectroscopy (EIS) is a method for analyzing the electrochemical reaction kinetics of electrode materials. EIS was also performed for the above electrodes (C/KOH-3, C/KOH/S-3:5, C/KOH/S-3:3-400 and C/KOH/S-3:3-700). The resulting Nyquist plot is shown in Figure 8c. In the low-frequency region, the impedance curve is more vertical, which is characteristic of capacitive behavior [80]. The intercept of Z’ at high frequency, which represents the equivalent series resistance (ESR), includes the ionic resistance of the electrolyte, the intrinsic resistance of the electrode, and the contact resistance of the active material. The arc diameter of the semicircle represents the charge transfer resistance (R_ct_) at the electrode/electrolyte interface. Within the frequency range from 100 KHz to 10 MHz, the C/KOH-3, C/KOH/S-3:5, C/KOH/S-3:3-400 and C/KOH/S-3:3-700 electrodes exhibit ESR resistance values of 1.85 Ω, 1.8 Ω, 2.1 Ω, and 1.4 Ω, respectively, and there is no obvious semicircular arc in the curves of the electrodes in the middle- to high-frequency regions, indicating that the interface impedance between the electrode surface and the electrolyte is very low [81]. The figure shows that the C/KOH/S-3:3-700 electrode has the lowest ESR and the lowest R_ct_ value, which can probably be attributed to the high degree of graphitization, high specific surface area, and presence of more surface-active species (doped) of the electrode material (AC) caused by the dopants [82].

The stability test of C/KOH-3 and C/KOH/S-3:3-700 electrodes is shown in Figure 9a. The specific capacitance values were compared after 7000 cycles of charge/discharge at a current density of 1 A g^−1^. Within 1000 cycles, the specific capacitance of C/KOH/S-3:3-700 decreased faster than that of C/KOH-3, which may be due to the main pore diameter of the C/KOH/S-3:3-700 electrode material being larger than that of the C/KOH-3 material. Therefore, the structural strength of the C/KOH/S-3:3-700 electrode material is low, and the structure can easily collapse during the charging and discharging process, resulting in a significant retention rate decrease in terms of capacitance, but this gradually became stable after 1000 cycles. However, the electrode was examined by SEM, and the electrode morphology was compared before and after the 1000 charge/discharge cycles test, and only very few cracks could be found (Appendix A). The C/KOH-3 and C/KOH/S-3:3-700 electrodes were charged/discharged for 7000 cycles, and their initial capacitance retention rates were 98.7% and 95.6%, respectively, as shown in Figure 9b. Thus, this capacitor material made from coconut shell biomass waste from Taiwan as a carbon source is able to significantly improve the electrochemical characteristics of supercapacitors and demonstrate excellent energy storage performance.

## 4. Conclusions

In this study, waste coconut husk biomass materials were used as a carbon source, KOH was used as an activator, (NH_4_)_2_S_2_O_8_ was added as N, S and O as a heteroatom doping source, and chemical activation was performed at high temperature to successfully prepare N-, S- and O-co-doped hierarchical porous activated carbon. The as-obtained carbon material was fabricated as electrodes and the electrochemical properties were characterized. The chemical activation of KOH can effectively promote the generation of hierarchical porous carbon materials, significantly increasing the specific surface area of the activated carbon materials and resulting in a high specific surface area of 1412.8 m^2^·g^−1^. Heat treatment at a high temperature of 700 °C is beneficial to the “graphitization” of carbon materials, and can improve the capacitance characteristics of electrode prepared. The prepared hierarchical porous heteroatom-co-doped activated carbon material contained 4.76%, 6.78% and 3.88% of S, O and N dopant, respectively; therefore, its electrochemical behavior had the characteristics of both pseudo-capacitance and electric double-layer capacitor, and its capacitance value reached 184 F g^−1^. At a current density of 1 A g^−1^, after 7000 cycles of charging and discharging stability tests, the material still retained 95.6% of the initial capacitance. Thus, coconut shell biomass waste in Taiwan can be used for the preparation of activated carbon, in order to significantly improve the electrochemical characteristics of supercapacitors.

## Data Availability

Not applicable.

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
