# Peer review of "Hierarchical Porous Heteroatoms—Co-Doped Activated Carbon Synthesized from Coconut Shell and Its Application for Supercapacitors"

_nanomaterials, 2022, doi:10.3390/nano12193504_

Round 1
Reviewer 1 Report
In the manuscript entitled "Hierarchical Porous Heteroatoms — Codoped Activated Carbon Synthesized from Coconut Shell and its Application for Supercapacitor", authors have prepared activated carbons using simultaneously activating agent and dopant (potassium hydroxide and ammonium persulfate). Coconut shell was applied as a carbonaceous precursor. The prepared porous carbons were tested as a material for supercapacitor electrodes.
The design of the experiment is good. The interpretation of the results needs improvement. However, the main claim to this manuscript is the novelty of the study. The authors did not clearly indicate the novelty of this study. Using coconut shells to make activated carbon does not seem like a very new concept. Coconut shell is a well-known carbon precursor for the production of porous carbon. The application of coconut-derived porous carbons as an energy storage material has also been studied quite a long time and extensively.
Coconut Shell-Derived Activated Carbon for High-Performance Solid-State Supercapacitors
https://doi.org/10.3390/en14154546
Activated carbons derived from coconut shells as high energy density cathode material for Li-ion capacitors
https://doi.org/10.1038/srep03002
Coconut-Shell-Based Porous Carbons with a Tunable Micro/Mesopore Ratio for High-Performance Supercapacitors
https://doi.org/10.1021/ef3009234
For doping of activated carbon with nitrogen and sulfur atoms during its synthesis, an additional agent (containing sulfur and/or nitrogen) is added to the main activating agent. This idea is also covered quite extensively in the literature.
More Sustainable Chemical Activation Strategies for the Production of Porous Carbons
https://doi.org/10.1002/cssc.202001838
Straightforward synthesis of Sulfur/N,S-codoped carbon cathodes for Lithium-Sulfur batteries
https://doi.org/10.1038/s41598-020-61583-1
Nitrogen- and Sulfur-Doped Carbon Obtained from Direct Hydrothermal Carbonization of Cellulose and Ammonium Sulfate for Supercapacitor Applications
https://doi.org/10.1021/acssuschemeng.0c05520
3-Dimensional hierarchical porous activated carbon derived from coconut fibers with high-rate performance for symmetric supercapacitors
https://doi.org/10.1016/j.matdes.2016.08.070
Influence of Surface Chemistry on the Electrochemical Performance of Biomass-Derived Carbon Electrodes for its Use as Supercapacitors
https://doi.org/10.3390/ma12152458
High Electrochemical Performance from Oxygen Functional Groups Containing Porous Activated Carbon Electrode of Supercapacitors
https://doi.org/10.3390/ma11122455
In addition, the manuscript contains other shortcomings. The main shortcomings of the manuscript are as follows:
1) In my opinion it is incorrect to write in the Abstract that "amount of dopant source and carbonization temperature were investigated to reveal the influences on the pore size distribution" while the figures show the pore size distribution only down to ~1.7 nm. Such activated materials are more often microporous with an average pore size of about 0.5 nm.
2) It is necessary to write what supercapacitors are and why they are important. And then discuss the background and relevance of materials for supercapacitors. I believe that this will improve the understanding of the article.
3) Why did the authors choose a temperature of 700 for activation?
4) When discussing the morphology of the activated samples, the authors write that no pores form on sample C/KOH-1. This is a frequent mistake when describing SEM images. The scale of such photographs does not allow us to clearly state the presence or absence of porosity, since activated samples have numerous micropores. The spongy structure of the samples shown in the micrographs indicates successful activation. Areas with a diameter of 200 nm (approximately equal to the scale bar) can probably be called caverns.
5) A large miss is that the authors do not give a size distribution for the entire range of pores. As can be seen from the figures, the peak responsible for the volume of pores depending on their diameter is in the zone of microporosity. To characterize the porosity of the obtained samples, the authors should give the pore size distribution down to 0.4 nm.
6) Table 2 contains the same labels for doped samples.
7) The interpretation of the results of XPS analysis raises questions. The fitting of the peaks for the C1s region looks a little strange for this kind of material. Despite the presence of nitrogen in the porous carbon composition, it is hardly possible to obtain such an intense peak as indicated by the authors in Fig. 5b. By the way, they do not discuss this region at all, limiting themselves to mentioning only the elements. The attribution of the peaks should be reconsidered, supported by references to similar materials. In addition, we can see that the authors have taken the fitting of the peaks lightly. This is evidenced by the absence of a unified design of the figures. In Fig. 6e, the authors very carelessly drew the background line. The authors should bring all drawings to the same style and type of background line (Shirley). The drawings should show raw spectrum, fitted peaks, summed spectrum, and peak fitting error.
8) Obviously, if the authors talk about the influence of dopant (ammonium persulfate) on the composition of the obtained samples, then in Table 3 it is necessary to give the chemical composition of the underdoped activated sample. This will show how important the role of dopant is.
However, there are other less significant defects in the manuscript. For instance, the sentence “Thus, the specific surface area, pore structure, surface functional groups and conductivity of activated carbon are important properties that determine the capacitance performance of carbon materials [11]” is a repeat of the first sentence in the Introduction. Another but not the least flaw is the very frequent mention by authors that SSA and porosity are very important. I think it is enough to mention it in the introduction. Having corrected the previous major shortcomings, the other smaller ones will be easy to correct. The authors should improve the English of this manuscript.
Reviewer 2 Report
The authors synthesized and characterized an electrode material: hierarchical porous activated carbon from coconut husk biomass, and N, S, and O heteroatoms codoped activated carbon electrode manufactured at 700 °C for electrochemical characteristics. The performance of as-prepared electrode materials is acceptable. Before it can be further considered for publication in nanomaterials, this manuscript needs to be improved. Some points are suggested as follows:
1. The numbering in all figures in the manuscript is mentioned two times, and the authors are advised to write only at the top right-side corner of the figure.
2. The data presentation and explanation can be refernce from Chemosphere, Volume 303, Part 3, September 2022, 135290 (10.1016/j.chemosphere.2022.135290 ) with proper citation.
3. The arrangement of the figure in figure 5 does not seem reasonable, so please arrange the figure in a complete pattern. Furthermore, maintain uniformity in all figures.
4. The ID/IG of sample C/KOH/S-3:3-700 is very low than all prepared samples. What is the primary reason behind this? And the authors are advised to write more briefly about the effect of the degree of graphitization on electrochemical performance in the manuscript.
5. XRD data is available in the manuscript. The authors are advised to include the XRD pattern of all samples with explanations in the manuscript.
6. The charging/discharging mechanism is not mentioned clearly in the manuscript, and the authors are advised to explain it more clearly.
7. It is advised to supplement the relevant data of the cyclic GCD stability test after 10,00 cycles of electrode materials.
8. The quantity of active materials used in developing the electrodes is not mentioned. So, the authors are advised to mention this.
9. The authors may need to cite more relevant literature to enrich their introduction, such as Chemical Engineering Journal, 450, Part 4, 2022, 138363 and Carbon, Volume 201, 5 January 2023, Pages 12-23
Reviewer 3 Report
The manuscript is very well written however I have a few suggestions for possible improvement of the article
[1] Compare the performance and processing method with CNC supercapacitors
A few references:
Yuan, H., Pan, H., Meng, X., Zhu, C., Liu, S., Chen, Z., ... & Zhu, S. (2019). Assembly of MnO/CNC/rGO fibers from colloidal liquid crystal for flexible supercapacitors via a continuous one-process method. Nanotechnology, 30(46), 465702.
Su, D., Pan, L., Fu, X., & Ma, H. (2015). Facile synthesis of CNC–MnO2 hybrid as a supercapacitor electrode. Applied Surface Science, 324, 349-354.
Durairaj, A., Maruthapandi, M., Saravanan, A., Luong, J. H., & Gedanken, A. (2022). Cellulose Nanocrystals (CNC)-Based Functional Materials for Supercapacitor Applications. Nanomaterials, 12(11), 1828.
[2] comment on adaptability of process for large scale up
[3] comment on sustainability of the process
Round 2
Reviewer 1 Report
The authors have answered my questions and comments quite well and corrected some shortcomings. However, I would advise the authors once again to pay attention to several points.
The authors claim the following definition of the novelty of their research: “The novelty of this study is the utilization of the chemical composition of ammonium persulfate, which is also a dopant source of N, S and O heteroatoms”. Here are some publications where ammonium persulfate is used to modify activated carbons.
0.1021/la00011a035
10.1016/j.jece.2016.10.028
10.1007/s10008-016-3452-8
10.3390/ma12152458
In my opinion, the authors should reformulate their statement and be more precise in defining novelty.
Other remarks:
1) In Introduction. “These carbon materials can be used to prepare supercapacitors with stable cycle properties…” Why “these”? The authors talk about supercapacitors, but not about carbon materials. This sentence seems strange to me, as if you were already talking about carbons.
2) “When the operating temperature was higher than 750 °C, because of the large amount of oxygen generated during the reaction process, the coconut shell cannot smoothly be turned into activated carbon.” What does it mean? This is not a very clear reason for the choice of temperature. What exactly is happening to the material?
3) “and treated in a manner the same as the above undoped AC at a temperature of 400-700 °C min-1 in order to”. These are units of heating rate, not temperature.
4) Unfortunately, the authors ignored my next point: “The attribution of the peaks should be reconsidered, supported by references to similar materials”.
I will allow myself to make some additional comments, which should be taken as advice to improve the description of their future studies.
1) I have doubts about the correctness of the fitting of the peaks on the XPS spectrum of C1s. The authors' persistent desire to give a peak for C-N could mislead the authors. I would advise the authors to read the guides on the interpretation of the spectra of carbon materials.
2) The problem with representing the pore size distribution over a wide range is clear. Our research group has exactly the same device. However, it can be used to conduct studies using the small steps necessary to measure microporosity. A software add-on is needed. Authors should look for it from colleagues or from the manufacturer.
Reviewer 2 Report
All the references are not listed in the reference section of the main article (Only 59 references are listed, but in the article, there is reference number up to 84).
In the answer section, Comment 9 related reference number 64 there is missing some of the initial authors' names. Re-write It.
Furthermore, the template use and the formatting of the article should be done appropriately, before final submission.
